# In Vivo Studies of Inoculated Plants and In Vitro Studies Utilizing Methanolic Extracts of Endophytic *Streptomyces* sp. Strain DBT34 Obtained from *Mirabilis jalapa* L. Exhibit ROS-Scavenging and Other Bioactive Properties

**DOI:** 10.3390/ijms21197364

**Published:** 2020-10-06

**Authors:** Ajit Kumar Passari, Vincent Vineeth Leo, Garima Singh, Loknath Samanta, Heera Ram, Chandra Nayak Siddaiah, Abeer Hashem, Al-Bandari Fahad Al-Arjani, Abdulaziz A. Alqarawi, Elsayed Fathi Abd_Allah, Bhim Pratap Singh

**Affiliations:** 1Department of Biotechnology, Mizoram University, Aizawl 796004, Mizoram, India; ajit.passari22@gmail.com (A.K.P.); vincentvineethleo@gmail.com (V.V.L.); loknathsamanta@gmail.com (L.S.); 2Departamento de Biología Molecular y Biotecnología, Instituto de Investigaciones Biomédicas, Universidad Nacional Autónoma de México (UNAM), Ciudad de Mexico 04510, Mexico; 3Department of Botany, Pachhunga University College, Aizawl 796004, Mizoram, India; garima.singh106@gmail.com; 4Department of Zoology, Jai Narain Vyas University, Jodhpur Rajasthan 342001, India; hr.zo@jnvu.edu.in; 5DOS in Biotechnology, University of Mysore, Manasagangotri, Mysore 570005, India; moonnayak@gmail.com; 6Botany and Microbiology Department, College of Science, King Saud University, P.O. Box. 2460, Riyadh 11451, Saudi Arabia; habeer@ksu.edu.sa (A.H.); aalarjani@ksu.edu.sa (A.-B.F.A.-A.); 7Mycology and Plant Disease Survey Department, Plant Pathology Research Institute, ARC, Giza 12511, Egypt; 8Plant Production Department, College of Food and Agricultural Sciences, King Saud University, P.O. Box. 2460, Riyadh 11451, Saudi Arabia; alqarawiwi@ksu.edu.sa (A.A.A.); efabdallah@gmail.com (E.F.A.); 9Department of Agriculture & Environmental Sciences (AES), National Institute of Food Technology Entrepreneurship & Management (NIFTEM), Sonepat 131028, Haryana, India

**Keywords:** *Streptomyces*, ROS, antioxidant, cytotoxicity, GC-MS, antidiabetic

## Abstract

Reactive oxygen species (ROS) and other free radicals cause oxidative damage in cells under biotic and abiotic stress. Endophytic microorganisms reside in the internal tissues of plants and contribute to the mitigation of such stresses by the production of antioxidant enzymes and compounds. We hypothesized that the endophytic actinobacterium *Streptomyces* sp. strain DBT34, which was previously demonstrated to have plant growth-promoting (PGP) and antimicrobial properties, may also have a role in protecting plants against several stresses through the production of antioxidants. The present study was designed to characterize catalase and superoxide dismutase (SOD), two enzymes involved in the detoxification of ROS, in methanolic extracts derived from six endophytic actinobacterial isolates obtained from the traditional medicinal plant *Mirabilis jalapa*. The results of a preliminary screen indicated that *Streptomyces* sp. strain DBT34 was the best overall strain and was therefore used in subsequent detailed analyses. A methanolic extract of DBT34 exhibited significant antioxidant potential in 1,1-diphenyl-2-picrylhydrazyl (DPPH) and 2,2′-azino-bis-3-ethylbenzthiazoline-6-sulphonic acid (ABTS) assays. The cytotoxicity of DBT34 against liver hepatocellular cells (HepG2) was also determined. Results indicated that methanolic extract of *Streptomyces* sp. strain DBT34 exhibited significant catalase and SOD-like activity with 158.21 U resulting in a 55.15% reduction in ROS. The IC_50_ values of a crude methanolic extract of strain DBT34 on DPPH radical scavenging and ABTS radical cation decolorization were 41.5 µg/mL and 47.8 µg/mL, respectively. Volatile compounds (VOC) were also detected in the methanolic extract of *Streptomyces* sp. strain DBT34 using GC-MS analysis to correlate their presence with bioactive potential. Treatments of rats with DBT34 extract and sitagliptin resulted in a significant (*p* ≤ 0.001) reduction in total cholesterol, LDL-cholesterol, and VLDL-cholesterol, relative to the vehicle control and a standard diabetic medicine. The pancreatic histoarchitecture of vehicle control rats exhibited a compact volume of isolated clusters of Langerhans cells surrounded by acinies with proper vaculation. An in-vivo study of *Streptomyces* sp. strain DBT34 on chickpea seedlings revealed an enhancement in its antioxidant potential as denoted by lower IC_50_ values for DPPH and ABTS radical scavenging activity under greenhouse conditions in relative comparison to control plants. Results of the study indicate that strain DBT34 provides a defense mechanism to its host through the production of antioxidant therapeutic agents that mitigate ROS in hosts subjected to biotic and abiotic stresses.

## 1. Introduction

Plants have close interactions with a wide array of microorganisms that colonize the rhizosphere, phyllosphere, and endosphere of plants. Endophytes are microorganisms that reside within plant tissue without causing disease and establish a synergistic affiliation with their host plants. Plants have developed an information transfer system with endophytic microorganisms that contributes to enhanced tolerance to stresses that induce the generation of reactive oxygen species (ROS), and the synthesis of plant growth-promoting substances [1,2,3]. When present in excessive amounts, common reactive oxygen species (ROS), such as O_2_^−^, H_2_O_2_and OH, can cause extensive oxidative damage to cells and generally have an adverse effect on cell metabolism. They are generated as a byproduct of normal metabolism, including respiration and photosynthesis [2,4].

Endophytic actinobacteria affiliated with medicinal plants have been shown to have the potential to inhibit or kill pathogenic bacteria, fungi, and viruses. Thus, they are considered as a significant source for the development of new antimicrobial products [3,5]. In our earlier studies, we demonstrated that endophytic bacteria could be a potent source for secondary metabolites with bioactive potential [6,7,8,9]. Additionally, bioactive compounds with cytotoxic and antioxidant properties produced by endophytic actinobacteria associated with plants have also been reported. Among these are endophytic actinobacteria, which exhibit cytotoxicity against several cancer cell lines [10,11]. Tanvir et al. [12], reported antioxidant activity in most of the endophytic actinobacteria (66.6%) that were recovered from different medicinal plants. Presently, researchers are continuing to search for novel actinobacteria with antioxidant properties for therapeutic use. *Mirabilis jalapa* L., a member of the *Nyctaginaceae*, is a traditional medicinal plant whose plant parts can be used to make a drink that is orally consumed 2–3 times a day (10–15 mL) for the treatment of kidney and urinary infections [13]. Rozina [14] reported that *M. jalapa* had several pharmacological functions, including antimicrobial, antimalarial, antioxidant, cytotoxicity, and antifungal properties.

In the previous study, six endophytic isolates (*Streptomyces* sp. strain DBT33; *Streptomyces* sp. strain DBT34; *Brevibacterium* sp. strain DBT35; *Streptomyces thermocarboxydus* strain DBT36; *Actinomycete* strain DBT37 and *Streptomyces* sp. strain DBT39) obtained from *M. jalapa* were tested for antimicrobial activities against three bacterial pathogens (*Pseudomonas aeruginosa*, *Staphylococcus aureus*, *Escherichia coli*, and *Candida albicans*). Isolate DBT35 showed significant antimicrobial activities against *S. aureus* (13.7 mm) and *P. aeruginosa* (10.1 mm), whereas BPSAC39 (10.6 mm) and BPSAC37 (9.6 mm) exhibited acute activities against *P. aeruginosa* and *C. albicans*, respectively. Although isolate DBT34 showed lower antimicrobial activities against all bacterial pathogens, isolate DBT34 showed strong antifungal activities against plant fungal pathogens *Rhizoctonia solani*, *Fusarium graminearum* and *Fusarium oxysporum*. In addition, isolate DBT34 showed maximum PGP activity in comparison to other isolates. The potent isolate DBT34 was used for an in-vivo plant growth promotion study under greenhouse conditions in an effort to enhance the growth of *Capsicum annuum* L. [6,7]. Notably, however, no systematic study has been conducted to understand the cytotoxicity, antioxidant, and antidiabetic potential of the endophytic actinobacteria associated with *M. jalapa*. Hence, we have selected the strain DBT34 to be evaluated for its antioxidant potential to alleviate the oxidative stress in their host plants by scavenging the ROS. Additionally, DBT34 was also selected to experimentally assess its anticancer ability against liver hepatocellular cancer cells (HepG2) lines and also antidiabetic capabilities. Also, a methanolic extract of isolate DBT34 was used to identify the volatile compounds (VOCs) that are potentially related to functional mitigation of stress in plants. The majority of researchers have reported antioxidant and antidiabetic potential from diverse bacteria, but there have been few reports from actinobacteria. Hence, we have added to evaluate an endophytic *Streptomyces* sp. isolate which has the potential for antioxidant, cytotoxicity, and antidiabetic capabilities. The results of this study provide in-depth information about the role of endophytic *Streptomyces* sp. in alleviated oxidative stress in their plant hosts through the production of antioxidant enzymes and compounds. 

## 2. Results

### 2.1. Catalase-Like and SOD-Like Activity

All of the methanolic extracts obtained from the different strains of *Streptomyces* sp. were evaluated for catalase-like activity based on their ability to degrade hydrogen peroxide (H_2_O_2_). Results indicated that extracts of six strains exhibited catalase-like activity, ranging from 19.40 U to 158.21 U at a concentration of 50 µg/mL. Among the tested strains, *Streptomyces* sp. strain DBT34 had the highest catalase-like activities at 158.21 U. Extracts of the six isolates were also evaluated for SOD-like activity. Results again revealed that the extracts of all six strains exhibited SOD-like activity ranging from 53.505% to 61.899% (Table 1). Isolate *Streptomyces* sp. strain DBT34 showed a free radical scavenging ability of 55.15%. 

### 2.2. Identification and Phylogenetic Analysis

The 16S rRNA sequence of strain DBT34 was compared with reference strains obtained from the GenBank database. Strain DBT34 was identified as *Streptomyces* sp. A phylogenetic tree was built using the neighbor-joining method. The estimated transition/transversion bias (R) was 1.38, and the overall pairwise mean distance was 0.081. The phylogenetic tree indicated that the 16S rRNA gene sequence of strain DBT34 was highly similar to *Streptomyces glauciniger* type strain CGMCC 41858 (99.51%), followed by *Streptomyces erringtonii* type strain I36 (98.92%), and *Streptomyces avellaneus* type strain NBRC13451 (97.35%). Notably, strain DBT34 formed a distinct clade with *Streptomyces rubrus* type strain Sp080513KE34 (96.96%), *Streptomyces baliensis* type strain ID030915 (96.96%), *Streptomyces malaysiensis* type strain NBRC16446 (96.96%), *Streptomyces sparsus* type strain YIM 90018 (96.86%), *Streptomyces spongiicola* type strain HNM0071 (96.7%), *Streptomyces wuyuanensis* type strain CGMCC 4.7042 (96.76%), *Streptomyces xanthocidicus* type strain NBRC13469 (96.75%), *Streptomyces tateyamensis* type strain DSM41969 (96.67%), *Streptomyces lactacystinicus* type strain OM6519 (96.67%), *Streptomyces coeruleoprunus* type strain NBRC15400 (96.67%), *Streptomyces griseoplanus* type strain NRRL B3064 (96.67%), and *Streptomyces cocklensis* type strain BK168 (96.57%) with a bootstrap value of 57% (Figure 1). 

### 2.3. Total Phenolics and Flavonoids 

The total phenolic content (TPC) in the methanolic extract of *Streptomyces* sp. strain DBT34 was as high as 94.21 µg of GAE per mg of DW, while the total flavonoid contact (TFC) was slightly higher (112.6 µg of QE per mg of DW) than the TPC content (Figure 2). Based on these data, it was concluded that *Streptomyces* sp. strain DBT34 might have antioxidant activity. 

### 2.4. Antioxidant Activity

Analysis of the methanolic extract of *Streptomyces* sp. strain DBT34 indicated that it had strong antioxidant activity as demonstrated by its DPPH and ABTS radical scavenging activity, producing IC_50_ values of 41.5 µg/mL and 47.8 µg/mL, respectively. These results indicate that low concentrations of the methanolic extract of strain DBT34 are capable of reducing DPPH and ABTS radicals, further demonstrating that strain DBT34 represents a potent source of antioxidant compounds that will degrade ROS (Figure 3). 

### 2.5. Cytotoxicity against Three Cancer Cell Lines

The cytotoxicity of a methanolic extract of strain DBT34 was screened against three cancer cell lines (HepG2, MCF7, AGS) at a concentration of 500 µg/mL. The preliminary screening indicated that the extract had a significant effect on the viability of HepG2 (human hepatocarcinoma) cells. Based on this result, the extract of strain DBT34 was tested for cytotoxicity at different concentrations (10, 25, 50, 100, and 250 µg/mL) against HepG2 cells using the MTT assay. Results indicated that the methanolic extract obtained from strain DBT34 exhibited the highest level of cytotoxicity against HepG2 cells at a low concentration, having an IC_50_ value of 49.2 μg/mL. Figure 4 illustrates the decrease in cell viability resulting from the induction of apoptosis induced by the methanolic extract of *Streptomyces* sp. DBT34. These results demonstrate that the crude methanolic extract of strain DBT34 may contain some bioactive compounds that are cytotoxic to HepG2 cancer cells. 

### 2.6. Gas Chromatography-Mass Spectroscopy (GC-MS) Analysis 

Volatile compounds (VOCs) present in the methanolic extract of DBT34 were identified by GC-MS. A total of 11 VOCs was predicted using the NIST library based on the peak area, molecular weight, and formula, the percentage of area and retention time (Table 2). The maximum percentage of peak area (17.427) was obtained for 4-Chlorobenzoic Acid, 4-Hexadecyl Ester, which exhibited a retention-time of 22.295. The minimum percentage peak area (5.629) was obtained for Benzeneacetic Acid, Alpha-Oxo, Trimethylsilyl Ester, which had a retention-time of 28.488. Among the 11 identified VOCs, two of the compounds, Cathinone (RT: 21.635) and Acetamide, 2-Amino (RT: 22.530) have been reported to possess antioxidant activity [15,16].

### 2.7. DPP-4 Inhibition In Vitro

The DBT34 extract and standard drugs exhibited significant (*p* ≤ 0.001) DPP-4 inhibition with values of 67.1% and 95.3%, respectively. The IC_50_ of the microbial extract was 0.87 µg/mL (Figure 5A,B).

### 2.8. Glucose Levels, Insulin, and HOMA Indices

The administration of a corticosteroid and high sucrose diet to the test rats caused a significant (*p* ≤ 0.001) elevation in glucose and insulin levels. The treatment of the test rats with either the methanolic extract of DBT34 or the standard drug sitagliptin had a significant impact on the elevation of glucose and insulin levels. Additionally, the HOMA-IR also significantly increased in the induced diabetic rats; however, treatment of the diabetic rats with either the DBT34 extract or sitagliptin caused a reduction in the HOMA-IR. More specifically, the HOMA-β and insulin sensitivity (IS) were lowered in the diabetic rats treated with either the DBT34 extract of sitagliptin and HOMA-β and IS were significantly (*p* ≤ 0.001) improved (Figure 6).

### 2.9. Alterations in the Lipid Profile of Treatment Groups

Diabetic rats treated with either the DBT34 microbial extract or the drug sitagliptin exhibited significant (*p* ≤ 0.001) reductions in total cholesterol, LDL-cholesterol, and VLDL-cholesterol in comparison to rats serving as the vehicle control and diabetic control. In contrast, HDL-cholesterol levels were not significantly altered (Figure 7).

### 2.10. Changes in Serum Antioxidant Levels

Rats that were induced to be diabetic exhibited significantly abnormal levels of SOD, catalase, GSH, and LPO. In contrast, diabetic rats that were treated with the microbial extract or sitagliptin showed significantly lower levels of LPO (lipid peroxidation). The levels of GSH, SOD, and catalase were also improved considerably in the treatment groups of rats (Figure 8).

### 2.11. Pancreas Histopathology

The histology of the pancreas of vehicle control rats was characterized by a compact volume of isolates of Langerhans surrounded by acini with proper vaculation. The islet of Langerhans exhibited a polymorphic cellular status (Figure 9A). In contrast, degenerative changes were observed in the islet of Langerhans in the pancreas of diabetic rats with areas of peripheral patches of cells in which β-cells were dominant and were characterized by pyknosis, necrosis, and a high level of apoptosis. The pancreas of diabetic rats also exhibited damage to blood vessels and diminished vascularization (Figure 9B). The pancreas of diabetic rats treated with the DBT34 microbial extract or sitagliptin showed regenerative activity in regard to the cellular morphology of islet of cells (Figure 9C,D).

### 2.12. Plant Growth Promotion in Chickpea Plants

#### Effect of *Streptomyces* sp. Strain DBT34 on Chickpea Plants In-Vivo and Antioxidant Activity in Derived Plant Extracts

The application of *Streptomyces* sp. strain DBT34 on chickpea plants and subsequent production of a methanolic extract from the treated and untreated plants resulted in IC_50_ values of 117.5 µg/mL (DPPH) and 114.0 µg/mL (ABTS) and IC_50_ values of 266.2 µg/mL (DPPH) and 169.3 µg/mL (ABTS) in untreated plants (Figure 10 and Appendix A). The lower IC_50_ values obtained for the methanolic extracts (µg/mL) of treated plants, relative to untreated plants, indicates that free radical scavenging in the treated plants was enhanced due to the presence of *Streptomyces* sp. strain DBT34 as an endophytic in the treated chickpea plants.

## 3. Discussion

Endophytes subsist inside host plant tissues without causing disease but protect the host by secreting several secondary metabolites, including antioxidants, which protect against ROS that are generated in response to stress conditions [2,21]. In their review, Hamilton et al. [2] discussed the role of endophytic fungi in mitigating ROS injury by increasing the antioxidant activity in the host plant.

Zhou et al. [22] reported that endophytic bacteria induce ROS directly by increasing the concentration of oxygen sesquiterpenoids content in the Chinese medicinal plant, *Atractylodes lancea*. No reports could be found, however, on endophytic Actinobacteria with ROS-scavenging potential. Recently, an association of endophytic Actinobacteria with medicinal plants from Northeastern India has been reported and suggested to represent a potential source for bioactive products, as well as antimicrobial and plant growth-promoting applications [6,7,8,9]. In the present study, *Streptomyces* sp. strain DBT34, isolated from *M. jalapa* L. growing in Mizoram, India was demonstrated to possess significant antioxidant activity, and cytotoxicity against cancer cell lines.

A phylogenetic tree was constructed using the neighbor-joining method to evaluate the similarity of *Streptomyces* sp. strain DBT34 with other species of *Streptomyces*. Results indicated that sequence of strain DBT34 was highly analogous to *Streptomyces glauciniger* type strain CGMCC 41858 (99.51%) followed by *Streptomyces erringtonii* type strain I36 (98.92%) and *Streptomyces avellaneus* type strain NBRC13451 (97.35%). In a similar analysis, Tan et al. [23] reported that *Streptomyces* sp. strain MUM256 exhibited a maximum 16S-rRNA gene sequence identity to *Streptomyces albidoflavus*DSM40455^T^ (99.7%) and *Streptomyces hydrogenans*NBRC13475^T^ (99.7%).

*Streptomyces* sp. strain DBT34 extract was evaluated for its ability to scavenge superoxide anion (O_2_^•−^)radicals, which can impact the synthesis of other ROS intermediates. SOD-like activity mitigates the accumulation of ROS intermediates by converting superoxide anion radicals to the lesser toxic entity, hydrogen peroxide. Hydrogen peroxide can then be neutralized by catalase, which turns into a neutral form [24]. Therefore, the present study evaluated catalase and SOD-like activity present in the methanolic extracts of DBT34 at a concentration of 50 µg extract/mL. These results are comparable to the level of inhibition reported for a microbial extract by Tan et al. [23]. In that study, the level of SOD-like activity was evaluated in methanolic extracts of *Streptomyces* sp. (MUM212). Their results indicated that SOD-like activity in methanolic extracts at a concentration range of 0.25 to 4 mg/mL exhibited a percentage of inhibition of superoxide radical formation ranging from 17 to 37%. Similarly, catalase-like activity was observed in methanolic extracts of *Gnaphalium polycaulon* Pers. plants [25].

Phenolic compounds are a significant group of antioxidants that also scavenge ROS. Cruz De Carvalho, [26] suggested that microorganisms produce ROS oxygen species as byproducts during the metabolism process. High levels of free radicals can induce oxidative stress and many other side effects, including the promotion of cancer [27]. It has been well established that free oxygen radicals can contribute to the development of a variety of diseases, including cancer, as well as neurodegenerative and cardiovascular diseases, which can be treated with the use of novel antioxidant compounds. In the present study, the total phenolic content in the methanolic extract of *Streptomyces* sp. strain DBT34 was estimated at 94.21 µg of GAE/mg of DW. These data represent a higher level than what was previously reported by Kaur et al. [28], who stated that an ethyl acetate extract of *Streptomyces* sp. strain OEAE (isolated from the soil) possessed a level of TPC of 84.3 mg of GAE/g DW. Lertcanawanichakul et al. [29] stated that the maximum amount of total phenolics (0.24 GAE ± 0.02 mg/g DW) was found in an ethyl acetate extract of *Streptomyces* sp. strain KB1-ET. Additionally, the maximum TFC in KB1-ET was 112.6 µg quercetin/mg DW. Our results represent the first report of the total phenolic content (TPC) and total flavonoid content (TFC) in a methanolic extract of *Streptomyces* sp. with the highest TPC and TFC exhibit the highest antioxidant activity [30].

Numerous techniques are available to assess the antioxidant capacity of organisms, among which the DPPH radical scavenging and ABTS radical cation decolorization assays represent rapid assays that are easy to perform. The free radical scavenging of the hydrogen atom donating antioxidant compound that allows for the conversion of hydrogen atoms or electrons to DPPH radicals results in the formation of a yellow-colored product, 1, 1-diphenyl-2-picryl hydrazine [21]. In the present study, strain DBT34 was found to exhibit significant antioxidant activity against DPPH free radicals with an IC_50_ value of 41.5 µg/mL. Kaur et al. [28] also reported that three *Streptomyces* sp. strains (TEAE, OCE, and TCE) isolated from soil exhibited the ability to scavenge DPPH radicals with IC_50_ values of 46.61 µg/mL, 51.88 µg/mL, and 89.03 µg/mL for the three strains, respectively. Tan et al. [31] reported that an ethanolic extract of *Streptomyces* sp. strain MUM256 had a potent antioxidant activity with IC_50_ values of 6.69 ± 0.83% and 12.08 ± 1.05%. The ABTS radical cation decolorization assay can also be used to assess the antioxidant potential of a test compound [32]. In our study, we found that strain DBT34 had a significant ability to scavenge ABTS radicals, having an IC_50_ value of 47.8 µg/mL. These data are in contrast to the results of Kaur et al. [28], who reported that the ABTS IC_50_ values of three *Streptomyces* sp. strains (TEAE, OCE, and TCE) were 121.51 µg/mL, 352.48 µg/mL, and 354.24 µg/mL, respectively.

We initially screened the crude methanolic extract of *Streptomyces* sp. strain DBT34 at a high concentration (500 µg/mL) for cytotoxicity against three cancer cell lines (MCF7, AGS and HepG2). The maximum anti-proliferative effect (cytotoxicity) was found against the HepG2 cell line with an IC_50_ concentration of 49.2 µg/mL. Khieu et al. [33] also reported that *Streptomyces* sp. strain HUST012 exhibited substantial cytotoxicity against HepG2 cells with an IC_50_ concentration of 41.63 µg/mL. Additionally, Lee et al. [34] reported that an ethanolic extract of *Streptomyces* sp. strain MJM 10778 exhibited cytotoxicity against HepG2 cell lines at an IC_50_ concentration of 264.7 µg/mL.

GC-MS analysis is an excellent method to detect and identify VOCs and has been used by several researchers to identify VOCs [35]. VOCs, including alcohol, ketones, esters, acetic acid groups, amides, and their derivatives, have been detected in genera of Actinobacteria [36]. In the present study, 11 volatile compounds were detected in the methanolic extract of *Streptomyces* sp. strain DBT34 using GC-MS analysis. Cathinone constituted 32.45% of the total amount of VOCs in strain DBT34. This compound was also reported by Dudai et al. [15], who stated that cathinone has antioxidant activity. The cathinone compound was reported in our isolate DBT34 and may be responsible for antioxidant activities. The volatile compound, acetamide, 2-Amino, was also detected in strain DBT34. Previous research has indicated that acetamide derivatives and their analogs have bioactive properties, such as antioxidant, anticancer, and anti-inflammatory activity [16,18]. The VOC, benzene acetic acid, 4-Methoxy-Alpha-[(Trimethylsilyl)Oxy]-Methyl compound, which is a derivative of 4-methoxyphenylacetic acid, as well as benzene acetic acid, 4-methoxy-p-methoxy-phenylacetic acid, was identified in the methanolic extract of *Streptomyces* sp. strain DBT34. Several researchers state that acetamide, 2-Amino and benzene acetic acid, 4-Methoxy-Alpha-[(Trimethylsilyl)Oxy]-Methyl compound showed anticancer activity as observed in our isolate DBT34 and could be responsible for cytotoxic potential against HepG2 cancer cell lines. A derivative of 4-Methoxybenzeneacetic acid has been reported to be neuroprotective, as well as possessing anticancer activity [19,20].

The historical usage of microbial secondary metabolites dates back to ancient civilizations in different food recipes and beverages for curing various ailments [37]. Using in-vitro, in-vivo and in-silico approaches, the present study evaluated the antidiabetic and antioxidant, properties of a microbial extract. In the present study, type 2 diabetes was induced in test rats by feeding them corticosteroids and a high sucrose diet, which has been reported to promote gluconeogenesis without adversely impacting carbohydrate metabolism [38]. Corticosteroids harm insulin signaling and functioning, resulting in insulin resistance [39]. Based on this effect, glucose homeostasis indices, including insulin resistance, reduced insulin sensitivity, and β-cell dysfunction, were assessed in test rats that were induced to be diabetic [40]. Insulin-resistant rats are unable to utilize glucose [41] correctly.

β-cells compensate for insulin resistance by increasing insulin secretion [42], which consequently results in a reduction of glucose uptake in body cells and promotion of hyperinsulinemia [40,43]. Additionally, alterations in the composition of saturated fatty acids in the plasma membrane result in reduced insulin signaling, which, in turn, results in a decrease in insulin sensitivity [44]. In our rat model system, the diabetic rats had abnormal lipid profiles, SOD-like, catalase-like, GSH activity, and lipid peroxidation.

Treatment of diabetic rats with the DBT34 microbial extract or sitagliptin mitigated the alterations in insulin resistance, β-cell function, and insulin sensitivity by reducing glucose and insulin levels. As reported in previous studies [45,46], the mitigation of altered metabolism in diabetic rats may have resulted from the effect of secondary metabolites present in the microbial extract on DPP-4 and other free radical scavenging activities. The decrease in DPP-4 accumulation may be due to the interaction between secondary metabolites in the microbial extract and DPP-4 at active sites [47].

Diabetic rats treated with the microbial extract or sitagliptin showed a reduction in the level of compounds in the lipid profile of diabetic rats that interfere with proper carbohydrate utilization, namely, total cholesterol, triglyceride, LDL-cholesterol, and VLDL-cholesterol that reduces free radical scavenging activity [48]. In support of the biochemical analyses, the histopathology of untreated diabetic rats indicated degenerative changes in the cellular morphology, all the way up to pyknosis, in the islet of the Langerhans in pancreatic tissues. It is likely that this may have resulted from reduced apoptosis as evidenced by the vacuolation and rearrangement of vascular tissues; which may result from reduced glucose uptake in pancreatic cells [49]. In contrast, diabetic rats treated with the microbial extract or sitagliptin exhibited reduced areas of vacuolation and improved vascularization in the islet of Langerhans in pancreatic tissues. The inhibition of DPP-4 accumulation and free radical scavenging activity has been reported to promote glucose availability and induce regenerative activity in damaged cells of insulin resistant tissues [50].

The antioxidant potential (ABTS and DPPH) present in 18 varieties of chickpea plants cultivated in different parts of the world, including four types from India, were characterized by Quintero-Soto et al. [51]. They reported antioxidant IC_50_ readings for DPPH scavenging ranging from 52 to 1640 μmol Trolox equivalents/100 g DW, while IC_50_ readings for ABTS ranged from 278 to 2417 μmol Trolox equivalents/100 g DW. These findings are accordance with the results obtained in the current study, where DPPH and ABTS scavenging activity in chickpea plants were assessed. Even though numerous in-vivo pot experiments studies have already been reported on the ability of endophytic Actinobacteria [52] or other bacteria [53] to promote growth in a variety of plants, only a few of these studies also assessed antioxidant potential. Recently, Wang et al. [54], studied the effect of endophytic *Streptomyces chartreusis* WZS021 on two sugarcane varieties under stress conditions and found that this Actinobacterium improved the stress tolerance of the host plant by regulating antioxidant levels. They postulated that the ROS production induced in host plants by the drought conditions was mitigated by the antioxidant activity of SOD, catalase, and peroxidase, which were enhanced by the endophytic strain, WZS021. Based on our data, we conclude that the endophytic *Streptomyces* sp., strain DBT34 has a significant level of antioxidant potential, along with substantial levels of TPC and TFC which may play a protective role in the host against abiotic and biotic stresses. Further in-depth studies are needed, however, to fully understand the mechanisms associated with the beneficial effects of endophytic Actinobacteria on their hosts, as well as the specific compounds and mechanisms responsible for the bioactive properties present in methanolic extracts of DBT34.

## 4. Materials and Methods

### 4.1. Isolation and Identification of Isolates of Actinobacteria

The six strains of Actinobacteria used in the present study were previously isolated and identified [6].

### 4.2. Extract Preparation

Six endophytic isolates of *Streptomyces* sp. (DBT 33, 34, 35, 36, 37, and 39) isolated from *M. jalapa* were evaluated for their antioxidant capacity and cytotoxicity to cancer cell lines. The same isolates were previously reported to possess significant antimicrobial and plant growth-promoting activity [6]. All six strains were individually streaked on starch casein agar (SCA) and tap water yeast extract agar (TWYE) media (obtained from Hi-media, India) and cultured at 28 °C for two weeks. Afterward, the Petri plates, which were completely covered with bacterial growth, were flooded with methanol solvent to obtain a crude extract as described by Abdalla [55].

### 4.3. Reactive Oxygen Species (ROS)-Scavenging Assays

#### Catalase and Superoxide Dismutase-(SOD)-Like Activity

The six crude extracts were dried using a rotary evaporator and diluted to 50 µg/mL in phosphate buffer (pH-8.0) and evaluated for catalase activity using a UV-Vis spectrophotometer. Hydrogen peroxide (H_2_O_2_) was added to the test to the solution, and catalase activity was measured as the decrease in H_2_O_2_. Absorbance was recorded at 240 nm at time intervals of 0 s; 30 s and 1 min. [25]. The superoxide anion scavenging activity or SOD like activity of DBT34 strain was evaluated using a UV-Vis spectrophotometer (Thermo scientific, Multiskan GO, MA, USA) to measure the formation of water-soluble formazan dye resulting from the enzymatic conversion of [2-(4-iodophenyl)-3-(4-nitrophenyl)-5-(2,4disulfophenyl)-2H-tetrazolium, monosodium salt] anion [23]. Briefly, a series of different concentrations of methanolic extract of *Streptomyces* sp. strain DBT34 were added into a 96-well plate along with [2-(4-iodophenyl)-3-(4-nitrophenyl)-5-(2,4disulfophenyl)-2H-tetrazolium, monosodium salt] anion. Then, the mixture was kept in the dark for 15 min to allow the formazan to form. Absorbance was subsequently recorded at 560 nm using a Multiscan UV/Vis spectrophotometer and used to calculate the level of SOD activity present in the extract.

### 4.4. 16S rRNA Amplification and Phylogenetic Analysis

All six isolates were identified using the amplification of 16S rRNA as previously described [6]. The best isolate, DBT34, based on the level of catalase and SOD activity, was taxonomically identified through phylogenetic analysis of the 16S sequence of DBT34 with type strains of *Streptomyces* present in the GenBank database using CLUSTAL-X software [56]. The sequences of the six isolates were also compared with the EzTaxon database [57], and a similarity percentage of 96.5–99.5% was revealed. A phylogenetic tree was built using the neighbor-joining method [58] with MEGA version 6.0 [59]. Tamura-3 parameter model (T92 + G + I) was selected based on BIC (2536.234) and AIC (2318.118) values with 1000 bootstrap replicates [58,60].

### 4.5. Phytochemical Analysis

#### Total Phenolic Content (TPC) and Total Flavonoid Content (TFC)

TPC in the methanolic extract of strain DBT34 was determined spectrophotometrically using the Folin–Ciocalteu (FC) reagent. Gallic acid was used to generate a standard curve in the range of 10–1000 mg/mL. A 200 µL volume of assay sample was prepared by combining 10 µL of sample extract with 90 µL FC reagent. This solution plus 100 µL of 15% of sodium carbonate were added to individual wells of 96-well plates. TPC was quantified as mg of Gallic acid equivalents (GAE) [61]. The total flavonoids content (TFC) in the methanolic extract of DBT34was determined using the aluminum colorimetric method. The standard curve of Quercetin solution in methanol was prepared with concentrations ranging from 0 to 500 µg/mL and absorbance was recorded at 420 nm with a UV-vis microplate spectrophotometer (MultiscanTM GO, Thermo Scientific, MA, USA). A total of 100 µL of sample extract (10–100 ug/mL) was mixed with 200 µL of 2% aluminum trichloride and used in the TFC assay. TFC was quantified as µg of Quercetin equivalents (QE) per mg of DW [62].

### 4.6. Determination of Antioxidant Potential

#### DPPH Radical Scavenging Assay and ABTS Radical Cation Decolorization Assay

The antioxidant potential of a methanolic extract of strain DBT34 was determined using a free radical DPPH scavenging assay and ABTS radical cation decolorization assay.

The ability of a methanol extract of DBT34 to scavenge the DPPH free radical was determined by using the stable 2, 2-diphenyl-1-picrylhydrazyl radical (DPPH). Briefly, for the DPPH scavenging assay, 100 µL of different concentrations (10–100 µg/mL) of the methanolic extract was placed in 96-well plate and mixed with 200 µL of DPPH solution (0.1 mM). The samples were kept in the dark for 39 min after which absorbance was recorded at 517 nm using a multi-scan UV/Vis spectrophotometer ((Multiscan^TM^ GO, Thermo Scientific, MA, USA). [63]. Ascorbic acid was used as positive control, methanol as a negative control and extract without DPPH was used as a blank. Results were expressed as a percentage reduction of DPPH absorption compared to control. In addition, the antioxidant potential of a methanolic extract of strain DBT34 was also determined with a 96-well plate method using an ABTS radical cation scavenging assay. Ascorbic acid, methanol, and extract without ABTS were used as positive, negative, and blank controls, respectively. The ABTS radical cation decolorization activity was calculated according to the method described by Re et al. [32].

### 4.7. Cytotoxicity

#### 4.7.1. Cell Lines and Culture Medium

Three cancer cell lines (HepG2, MCF7, and AGS) were obtained from NCCS, Pune, India. Cancer cells were grown in Dulbecco’s Modified Eagle’s medium (DMEM) media (Sigma-Alderich, USA) amended with 10% bovine serum, 100 µg/mL of streptomycin, 5 µg/mL of amphotericin B, and 100 IU/mL of penicillin and cultured at 37 °C in a humid environment and 5% atmospheric CO_2_. All of the analyses were carried out as described by Tan et al. [31].

#### 4.7.2. Cytotoxicity Assay

The MTT test was used to measure the cytotoxicity of the methanolic extract of DBT34 against three cancer cell lines. A total of 1.0 × 10^4^ cells per 100 μL of media were placed in 96-well plates. The plates were cultured in an incubator at 37 °C and 5% CO_2_ for 24 h. Initial cytotoxicity of a 500 μg/mL concentration of the methanolic extract was tested against all three cell lines and the results indicated that the extract had significant cytotoxicity against HepG2 cells. Based upon the preliminary results, the extract was diluted to different concentrations (10, 25, 50, 100, and 250 μg/mL), and their cytotoxicity against HepG2 cells was evaluated. Untreated cells served as a control. The methodology of the MTT assay was previously described by Tan et al. [31].

### 4.8. Determination of Volatile Organic Compounds (VOCs) Using Gas Chromatography-Mass Spectroscopy (GC/MS)

A methanolic extract of *Streptomyces* sp. strain DBT34 was used to identify VOCs using GC-MS. The GC-MS (Clarus 680) column was packed with Elite 5MS and had a size of 30 m × 0.25 mm ID × 250 μmdf. The compounds present in the extract were separated using Helium gas as a carrier at a constant flow of 1 mL/min. The instrument temperature was 260 °C when 1 μL of extract sample was injected. The oven temperature was set as follows: 60 °C (2 min), followed by 300 °C, increasing at a rate of 10 °C min^−1^. Samples were exposed to 300 °C for 6 min. The mass detector conditions were as follows: 240 °C for transfer line and ion source temperature, and the ionization mode electron impact was set at 70 eV, with a scan time and interval of 0.2 s and 0.1 s, respectively. Obtained spectra were compared with a local copy of the NIST (2008) library database.

### 4.9. Antidiabetic Assay

#### 4.9.1. Drugs and Chemicals

The commonly prescribed DPP-4 inhibitor drug, sitagliptin, was administered as a 25mg tablet by calculating dose (2.5 mg/kg/day), along with a tablet containing 4 mg dexamethasone. Both drugs were purchased from a local pharmacy.

#### 4.9.2. Animals

Healthy Wistar rats, weighing 170–190 gm, were maintained in polypropylene enclosures at a standard photoperiod (14 h light: 10 h dark) under controlled environmental conditions (26 ± 1 °C) and fed a standard laboratory feed (Gold Mohur feed, Hindustan Lever Limited, Mumbai, India) with water ad libitum. Animals were maintained as per the guidelines of the CPCSEA (Committee for Control and Supervision of Experiments on Animals), Ministry of Environment, Forest and Climate Change, Govt. of India (Reg. No.1646/GO/a/12/CPCSEA, valid up to 27.03.23). Approval date by CPCSEA was 29.03.2019.

#### 4.9.3. Induction of Type 2 Diabetes

Type 2 diabetes was induced by providing a high sucrose diet and the oral administration of dexamethasone (corticosteroid) at a calculated dose of 1.0 mg/Kg for three weeks, as described in previous studies [64,65,66]. Levels of glucose and insulin were analyzed from the first day to the end of the third week. Food intake, daily water consumption, and animal activity were also recorded on days 7, 14, and 21. The diabetic status of the animals was determined using homeostatic model assessment (HOMA) indices (HOMA-IR, HOMA β%, and HOMA sensitivity).

#### 4.9.4. In-Vitro DPP-4 Inhibition Assay

DPP-4 inhibition by the microbial extract and by sitagliptin was assessed based on the cleavage of Gly-Pro-p-nitroanilide by the DPP-4 enzyme resulting in the production of a stable chromophore. DPP-4 inhibition activity of the microbial extract and sitagliptin was assessed by determining the release of 4-nitroaniline from the assay mixture comprising 0.1 M Tris-HCl (pH 8.0) and 2 mMGly-Pro p-nitroanilide (substrate). The assay mixture was kept at 37 °C, and the reaction was terminated by the addition of sodium acetate buffer (pH 4.5). Absorbance was recorded at 405 nm using a UV-VIS Spectrophotometer [67,68]. Percent inhibition was calculated using the following formula:$$\%\;\mathrm{inhibition}=\frac{\mathrm{Absorbance}\;\mathrm{of}\;\mathrm{vehicle}\;\mathrm{control}-\mathrm{Absorbance}\;\mathrm{of}\;\mathrm{inhibitor}}{\mathrm{Absorbance}\;\mathrm{of}\;\mathrm{vehicle}\;\mathrm{Control}}\times 100$$

#### 4.9.5. Biochemical Assessments of Serum

##### Glucose, Insulin, HbA1C and Lipid Profile

Serum insulin [69], HbA1C [70], glucose [71], and the lipid profile including total cholesterol [72], HDL-cholesterol [73], and triglyceride [74] were assessed using commercially available kits. The level of LDL-cholesterol and VLDL cholesterol were calculated using Friedewald’s formula [75].
LDL-C = (TC) − (HDL-C) − (TG/5)

#### 4.9.6. Homeostatic Model Assessment (HOMA)

HOMA indices, including HOMA-IR (insulin resistance), HOMA-β (β-cell function), and insulin sensitivity (IS) were calculated using a formulated equation in Excel developed by the Diabetes Trials Unit (DTU), a registered UK Clinical Research Collaboration Clinical (UKCRC), and a registered Clinical Trials Unit [76,77].
$$\mathrm{HOMA}-\mathrm{IR}=\frac{\mathrm{Insulin}\;\left(U/L\right)\times \mathrm{Blood}\;\mathrm{Glucose}\left(\mathrm{mmol}/L\right)}{22.5}$$
$$\mathrm{HOMA}-\beta =\frac{20\times \mathrm{Insulin}\;\left(U/L\right)}{\mathrm{Blood}\;\mathrm{Glucose}\;\left(\mathrm{mmol}\;/L\right)}-3.5$$
$$\mathrm{Insulin}\;\mathrm{sensitivity}\;\left(\mathrm{IS}\right)=\frac{1}{\left[(\mathrm{Fasting}\;\mathrm{Insulin}\;\left(U/L\right)\times \mathrm{Log}\;\left(\mathrm{Fasting}\;\mathrm{glucose}\;\left(\mathrm{mmol}/L\right)\right)\right]}$$

#### 4.9.7. Serum Antioxidant Assay

Antioxidant enzymes, including catalase, glutathione (GSH), superoxide dismutase (SOD), and lipid peroxidase (LPO) in serum, were determined using standard methods [78,79,80,81,82]. LPO levels were determined based on thiobarbituric acid reactive substances (TBARS), and GSH was determined with the use of Ellman’s reagent (5,5′-dithiobis-2-nitrobenzoic acid (DTNB)), which produces a product that can be quantified spectrophotometrically at 412 nm. Catalase and SOD activity were measured using the Beauchamp and Fridovich methods [83,84].

#### 4.9.8. Histopathological Studies

Overnight fasted animals were autopsied under mild anesthesia, and the pancreas of the sacrificed rats were collected for histopathological examination. The collected pancreases were fixed in 10% formalin and subjected to gradual dehydration (alcohol series of 30%, 50%, 70%, 90%, and 100%), clearing, and paraffin embedding. Tissues were sectioned at 5 µ; sections were placed on glass slides, deparaffinized, and stained (H&E) using a standard protocol [85,86,87]. Clinical observations were made under a light microscope (Leica-DM RA, The Netherlands), and sections were photographed using a microscope-mounted camera.

### 4.10. Plant Growth Promotion Assay in Chickpea Plants (Cicer Arietinum)

#### Antioxidant Assay

To determine the effect of *Streptomyces* sp. strain DBT34 on the antioxidant activity in chickpea plants, an in-vivo experiment was conducted which involved treating chickpea seedlings with DBT-34 and letting the plants grow for four weeks. After that time, a methanolic extract (50–500 µg/mL) was prepared from the untreated four-week-old chickpea plants (control) and chickpea plants treated with *Streptomyces* sp. strain DBT34. The methanolic extract was used to conduct a DPPH radical scavenging assay and ABTS radical cation decolorization assay [32,63].

### 4.11. Statistical Analysis

A Tukey’s multiple range test (*p* < 0.05) was used to determine statistically significant differences between the experimental materials and the controls using SPSS software.

## Figures and Tables

**Figure 1 ijms-21-07364-f001:**
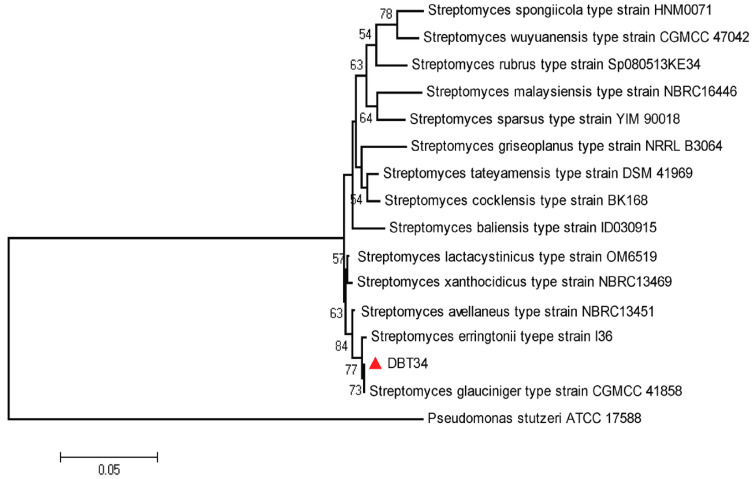
Phylogenetic relationship between *Streptomyces* sp., strain DBT34 and other strain types retrieved from the EZ-Taxon database, based on partial 16S rRNA gene sequences. The phylogenetic tree was constructed using the neighbor-joining method with the Tamura 3-parameter model with 1000 bootstrap replicates. Value mentioned at nodes is bootstrap values and our strain is mentioned against red triangle.

**Figure 2 ijms-21-07364-f002:**
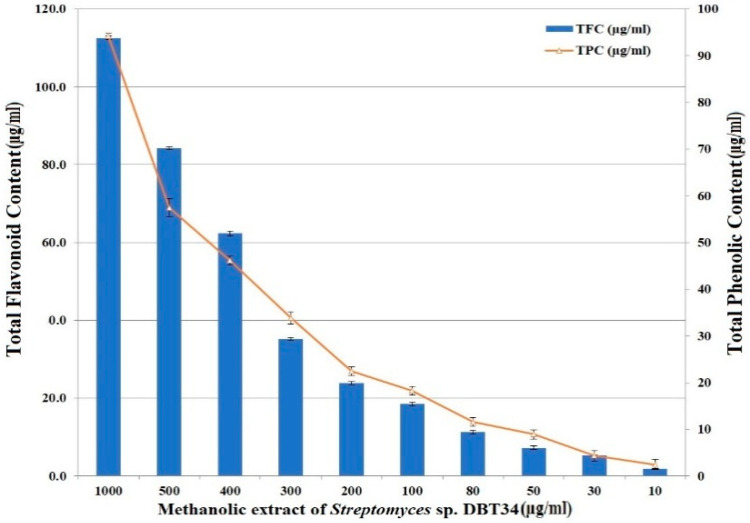
Total phenolic content and total flavanoid content in the methanolic extract of *Streptomyces* sp. strain DBT34. Data represents the mean ± SD.

**Figure 3 ijms-21-07364-f003:**
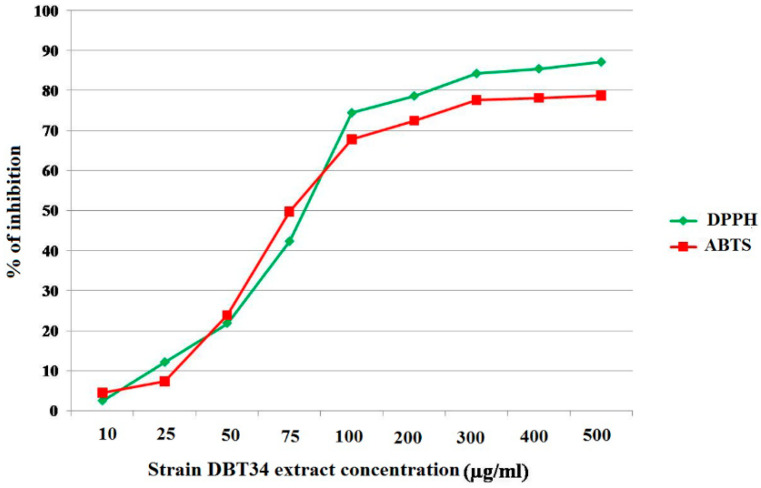
DPPH and ABTS radical scavenging activity of the methanolic extract of *Streptomyces* sp. strain DBT34.

**Figure 4 ijms-21-07364-f004:**
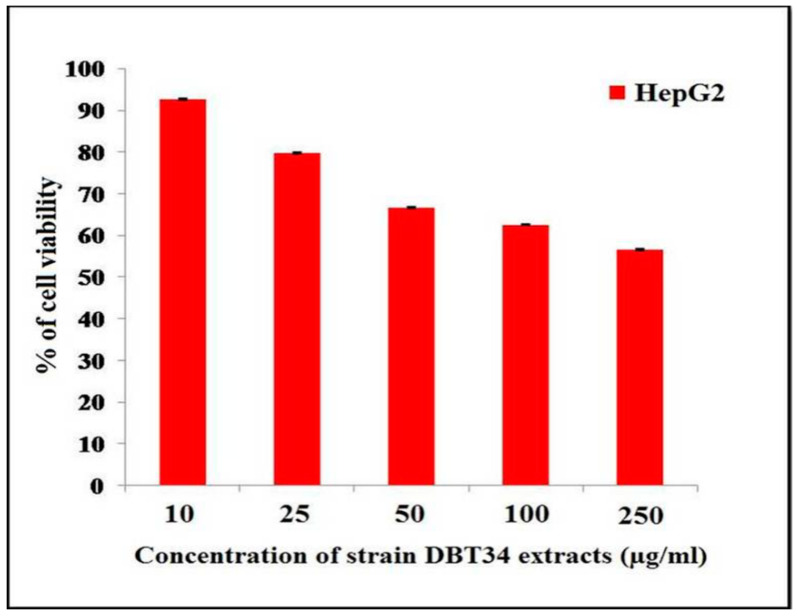
Effect of a 250 µg/mL methanolic extract of *Streptomyces* sp. strain DBT34 on the percentage of cell viability of HepG2 cancer cells. Each bar represents the mean ± SD.

**Figure 5 ijms-21-07364-f005:**
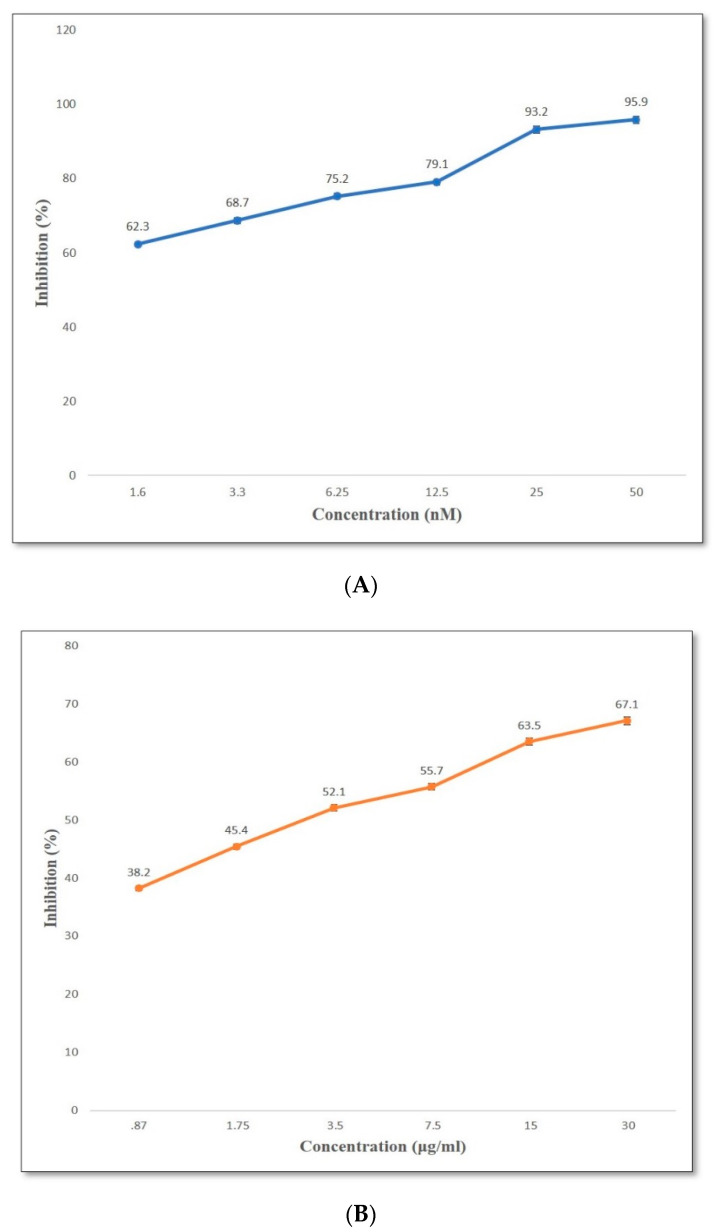
(**A**) Inhibition ofDPP-4 free radicals by an ascending concentration (nM) of sitagliptin. (**B**) Inhibition of DPP-4 free radicals by an ascending concentration (µg/mL) of DBT34 microbial extract.

**Figure 6 ijms-21-07364-f006:**
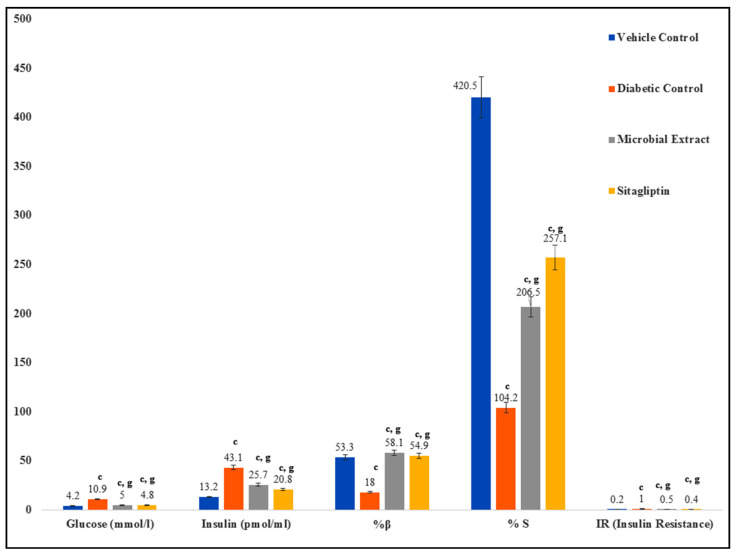
Glucose, insulin and HOMA indices in model system diabetic rats treated with DBT34 microbial extract or sitagliptin. Data are the mean ± S.E.M. (*n* = 5); c (*p* ≤ 0.001) and g (*p* ≤ 0.001) are significant.

**Figure 7 ijms-21-07364-f007:**
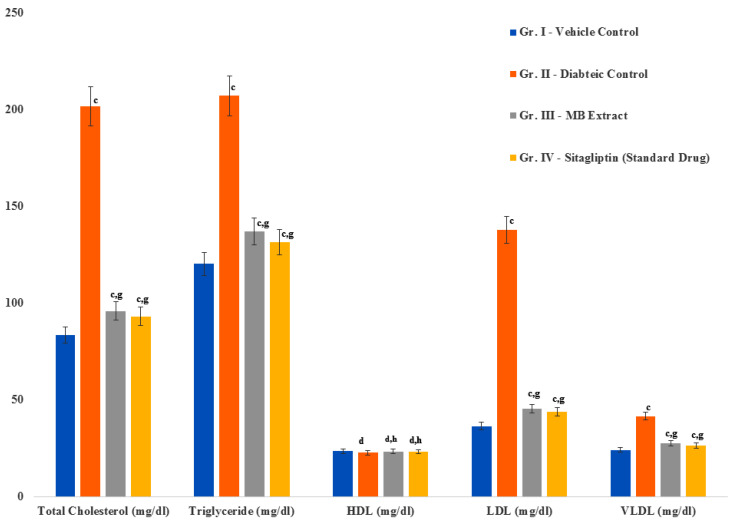
Alterations in the lipid profile of diabetic rats treated with DBT34 microbial extract and sitagliptin. Data are the ± S.E.M. (*n* = 5); Significant differences between treatment and control groups are indicated by the letters c (*p* ≤ 0.001); and d indicates a non-significant difference between the control and treatment groups. g (*p* ≤ 0.001); and h indicates a non-significant difference between the treatment groups and the hypercholesterolemic control group.

**Figure 8 ijms-21-07364-f008:**
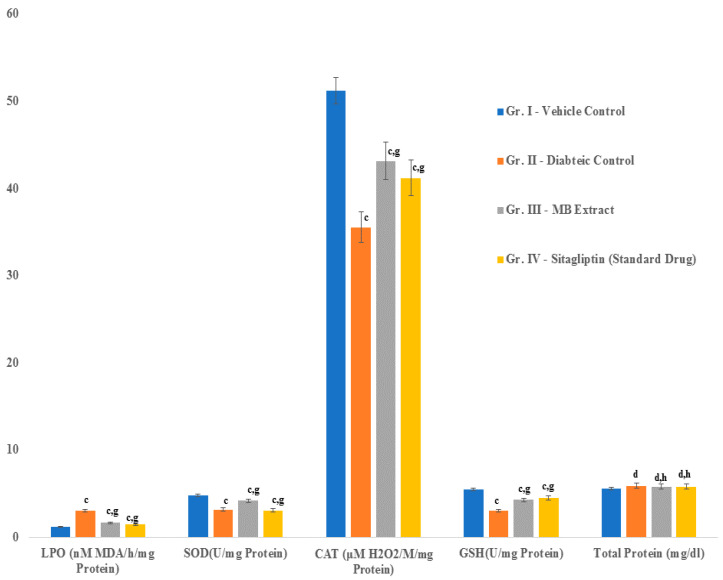
Changes in serum antioxidant levels in diabetic rats treated with DBT34 microbial extract or sitagliptin. Data are the mean ± S.E.M. (*n* = 5); letters indicate significant differences between the control and treatment groups c (*p* ≤ 0.001); and g (*p* ≤ 0.001) are significant. Whereas, d and h are non significant.

**Figure 9 ijms-21-07364-f009:**
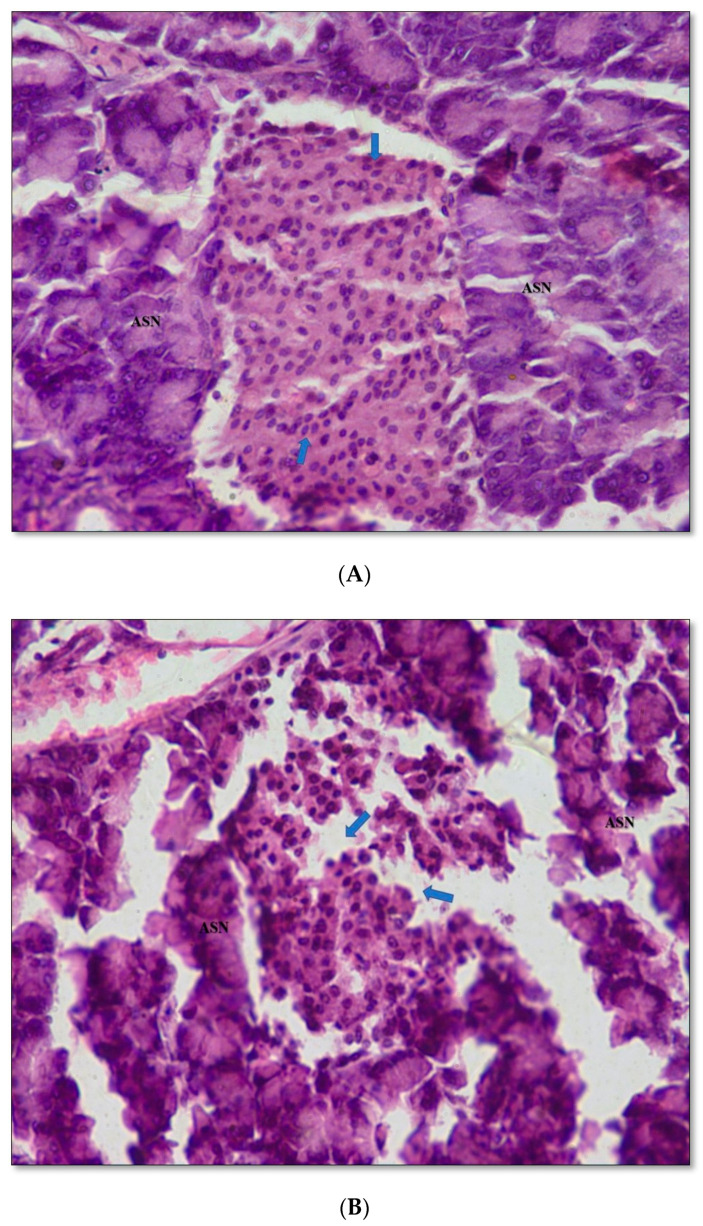
(**A**) Histopathology of a pancreas (**400X** HE) from an untreated non-diabetic rat (control): The arrows indicate a cellular mass of normal peripheral β-cells (PBC), distribution in islet (ISLT), and normal morphology of the arranged acini (ASN). (**B**) Histopathology of a T2DM control pancreas (**400X** HE) in untreated diabetic rats. The arrows indicate peripheral β-cells, rounded arrow indicates a congestion of RBC and the star indicates a vacuolated area in the pancreatic islet (ISLT) cells with abnormal appearance of the nucleus and abnormally arranged acini (ASN). (**C**) Histopathology of the pancreas of diabetic rats treated with an extract derived from DBT34 strain extract (**400X** HE): The arrows indicate a recovery of normal looking peripheral β-cells in islet (ISLT) and an increased cellular mass with normally arranged acini. (**D**) Histopathology of a pancreas of diabetic rats treated with sitagliptin (**400X** HE): The arrows indicate the regeneration of a damaged area of peripheral β-cells (PBC) and a compact arrangement of acini.

**Figure 10 ijms-21-07364-f010:**
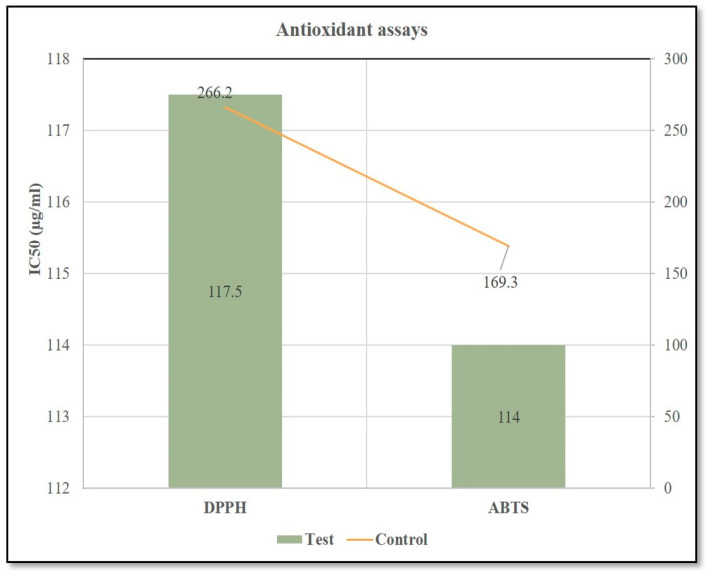
ABTS and DPPH radical scavenging activity methanolic extracts obtained from untreated chickpea plants and chickpea plants treated with bacterial cells of *Streptomyces* sp. strain DBT34. Cells were applied to the soil and plants were grown for 4 weeks prior to determining their antioxidant potential.

**Table 1 ijms-21-07364-t001:** Catalase and SOD like activity in a 50 µg/mL methanolic extract obtained from six different endophytic strains (DBT 33–39) of Actinobacteria (*Streptomyces* sp.).

ROS/Strain	DBT33	DBT34	DBT35	DBT36	DBT37	DBT39
Catalase-like activity (Unit)	64.31	158.21	49.27	19.49	65.5	39.4
SOD-like activity (% of inhibition)	53.51	55.15	59.84	61.89	56.47	61.75

**Table 2 ijms-21-07364-t002:** Identification ofvolatile compounds (VOCs) in a methanolic extract of *Streptomyce*s sp. strain DBT34.

Sl.No	Compound Name	Formula	MW	RT	Height	Area %	Norm %	Structure	Activities & References
**DBT34**	
1	ETHYL 6-[P-Chlorophenacylamino]-4-[[Diphenylmethyl]Amino]-5-Nitro-2	C_29_H_26_O_5_N_5_Cl	559	21.390	1,814,810	6.514	37.38	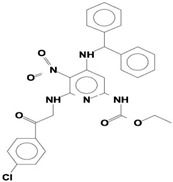	-
2	Cathinone	C_9_H_11_ON	149	21.635	1,398,351	5.655	32.45	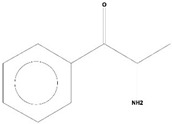	Antioxidant activity [17]
3	4-Chlorobenzoic Acid, 4-Hexadecyl Ester	C_23_H_37_O_2_Cl	380	22.295	2,297,526	17.427	100.00	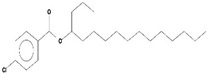	-
4	2,3-Dihydroxystearic Acid	C_18_H_36_O_4_	316	22.445	1,336,178	9.799	56.23	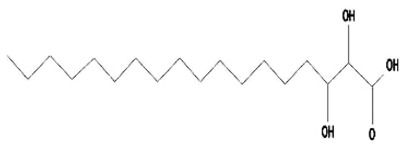	-
5	Acetamide, 2-Amino	C_2_H_6_ON_2_	74	22.530	1,669,673	7.104	40.76	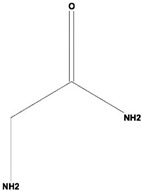	Antioxidant activity [16] & anticancer activity [18]
6	1,4-Cyclohexadiene, 1,3,6 Tris(Trimethylsilyl)	C_15_H_32_Si_3_	296	23.171	1,601,816	11.259	64.61	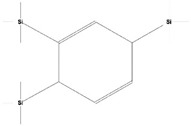	-
7	Benzeneethanamine, *N*-[(Pentafluorophenyl)Methylene]-Beta,4-Bis [(trimethylsilyl)oxy]	C_21_H_26_O_2_NF_5_Si_2_	475	28.408	1,640,921	10.878	62.42	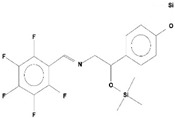	-
8	Benzeneacetic Acid, Alpha-Oxo Trimethylsilyl Ester	C_11_H_14_O_3_Si	222	28.488	2,008,596	5.629	32.30	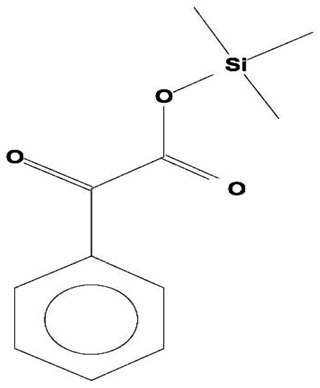	
9	Benzeneacetic Acid, 4-Methoxy-Alpha-[(Trimethylsilyl)Oxy]- Methyl	C_13_H_20_O_4_Si	268	28.688	1,468,754	5.683	32.61	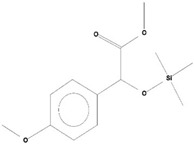	Anticancer activity [19]; Neuroprotective activity [20]
10	1,2-Benzenedimethanethiol, S-Trimethylsilyl	C_11_H_18_S_2_Si	242	28.793	1,568,043	6.188	35.51	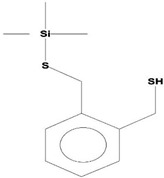	
11	Trimethyl[4-(1,1,3,3,-Tetramethylbutyl)Phenoxy]Silane	C_17_H_30_OSi	278	29.698	1,952,357	7.912	45.40	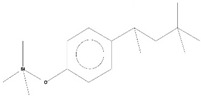

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
