# Peer review of "In Vivo Studies of Inoculated Plants and In Vitro Studies Utilizing Methanolic Extracts of Endophytic Streptomyces sp. Strain DBT34 Obtained from Mirabilis jalapa L. Exhibit ROS-Scavenging and Other Bioactive Properties"

_ijms, 2020, doi:10.3390/ijms21197364_

Round 1
Reviewer 1 Report
The article “In vivo studies of inoculated plants and in vitro studies utilizing methanolic extracts of endophytic Streptomyces sp. strain DBT34 obtained from Mirabilis jalapa L. exhibit ROS scavenging and other bioactive properties” provides investigation of antioxidant potential of methanolic extracts derived from six endophytic actinobacterial isolates obtained from Mirabilis jalapa.
The manuscript contains mix of data that are not correlated, however, there is not clear the aim of study (mosquitocidal activity along with antidiabetic and antioxidant activities)
It must be clear, but specific Introduction. In this manuscript, the introduction is too general and too long. Then it must be present aim or/and task, why these research was conduced. There is not aim.
English must be checked by native English speaker
References are tool old, must be updated
which compounds are responsible for HepG2 activity?
Work is too preliminary, just antioxidant, antidiabetic and cytotoxic activities, along with mosquitocidal properties with no mechanistic studies
The compounds responsible for activities should be reported.
Author Response
The authors are very much thankful to the reviewers and editor for their valuable comments. We have revised the manuscript as per your suggestions and incorporated the correction in the revised manuscript. Please find below the answers and rebuttals of the queries raised by editor/ reviewers.
Reviewer’s comment: 1
The manuscript contains mix of data that are not correlated, however, there is not clear the aim of study (mosquitocidal activity along with antidiabetic and antioxidant activities)
Author’s response: 1
Thank you for the reviewer's comments. The authors revised the aim of the study in the introduction section. We agree with the reviewer comments. There is no correlation between mosquitocidal activity, along with antidiabetic and antioxidant activities. Hence, we have removed the mosquitocidal analysis in the revised manuscript.
Reviewer’s comment: 2
It must be clear, but specific Introduction. In this manuscript, the introduction is too general and too long. Then it must be present aim or/and task, why these research was conduced. There is not aim.
Author’s response: 2
The authors have clearly mentioned the aim of the study and why this research is conducted in the revised manuscript. Also, the authors have changed the introduction section in the revised manuscript as per reviewers instructions.
Reviewer’s comment: 3
English must be checked by native English speaker
Author’s response: 3
The authors have made the English correction by native English speaker. We have done the editing of the manuscript again with a professional service (certificate attached).
Reviewer’s comment: 4
References are tool old, must be updated
Author’s response: 4
The authors have updated the reference in the revised manuscript as suggested by reviewer 1.
Reviewer’s comment: 5
Which compounds are responsible for HepG2 activity?
Author’s response: 5
We have identified 11 volatile compounds (VOCs) in methanolic extract of strain DBT34 using GC-MS analysis. Among them, two compounds Acetamide, 2-Amino and Benzeneacetic Acid, 4-Methoxy-Alpha-[(Trimethylsilyl)Oxy]- Methyl) which showed anticancer activity based on the literature was found in our isolates. Hence, the authors expecting that these two compounds might be responsible for HepG2 activity.
Reviewer’s comment: 6
Work is too preliminary, just antioxidant, antidiabetic and cytotoxic activities, along with mosquitocidal properties with no mechanistic studies
Author’s response: 6
Thank you for the reviewer comments. The authors decided to screen how actinobacteria isolate could be useful for antioxidant, antidiabetic and cytotoxic potential. We have seen that very less information available in actinobacteria. Hence, we have targeted to perform the analysis. We agree with reviewer comments. Further, mechanistic studies will be carried out in Streptomyces sp. to investigate which pathway or genes are responsible for antioxidant, antidiabetic and cytotoxic activities. The mosquitocidal analysis part has been removed in the revised manuscript.
Reviewer’s comment: 7
The compounds responsible for activities should be reported.
Author’s response: 7
The authors have been included compound responsible for activities in the discussion section as per the reviewer's request.
Reviewer 2 Report
The manuscript describes the characterization of methanolic extracts of Streptomyces DBT34 regarding it antioxidant activities. I have some concerns regarding the manuscript that prevents me from accepting it for publication in it present form.
- How DBT34 strain (and others strains) were obtained should be addressed in the introduction. There is no indication of the previous study in the introduction.
- I seriously doubt that catalase and sod activities were measured. Are the catalase and SOD enzymes functional after a methanolic extract. It seems that ros scavenging properties of the extract were analyzed instead, not enzyme activity. Do the authors have some evidence that there are functional catalase and sod enzymes in the extract? Performing a native page stained for catalase/sod activity it’s needed to show the presence of the enzymes
- the discussion section is too extended and includes large portions that are “state of the art” and not discussion. This make the discussion difficult to follow.
- what was the criteria for selecting the sequences from reference strains included in the phylogenetic tree? Information is missing.
- It is said that “Among all of the tested isolates, Streptomyces sp. strain DBT34 again showed the maximum ability to scavenge free radicals, with a scavenging ability of 55.15%.”. However in the data that is presented 4 out of the 6 tested extracts showed higher activity.
Author Response
Authors responses to reviewer(s) comments
The authors are very much thankful to the reviewers and editor for their valuable comments. We have revised the manuscript as per your suggestions and incorporated the correction in the revised manuscript. Please find below the answers and rebuttals of the queries raised by editor/ reviewers.
Reviewer #2
Reviewer’s comment: 1
How DBT34 strain (and others strains) were obtained should be addressed in the introduction. There is no indication of the previous study in the introduction
Author’s response: 1
We are thankful the suggestion of the esteemed reviewer. The authors have incorporated the previous study in the revised manuscript.
Reviewer’s comment: 2
I seriously doubt that catalase and sod activities were measured. Are the catalase and SOD enzymes functional after a methanolic extract. It seems that ros scavenging properties of the extract were analyzed instead, not enzyme activity. Do the authors have some evidence that there are functional catalase and sod enzymes in the extract? Performing a native page stained for catalase/sod activity it’s needed to show the presence of the enzymes.
Author’s response: 2
Thank you for the reviewer comments. Tan et al. (2017) reported that methanolic extract of Streptomyces sp. MUM212 extract exhibited significant SOD-like activity measuring from 17.78 ± 3.85 to 37.47 ± 1.79% at concentrations ranging from 0.25 to 4 mg/mL. Tan et al. suggested that MUM212 extract has the capability to scavenge the O2∙- produced from the hypoxanthine-xanthine oxidase system as reflected by the decrease in absorbance of the yellow water-soluble WST formazan which was formed upon reduction by O2∙-. Moreover, Shanmugapriya et al. (2017) reported that methanolic extracts of Gnaphalium polycaulon Pers showed maximum amount of catalase activity with 0.74 U/mg. Shanmugapriya et al. reported that SOD generates H2O2 as a more toxic product to the cells and required catalase or peroxidases to scavenge free radicals. An increase in catalase or peroxidase is essential for beneficial antioxidant effect from increase superoxide dismutase activity. Hence, we have suggested that catalase and SOD enzymes are functional after a methanolic extract. Similar information reported by Tan et al. (2017) and Shanmugapriya et al. (2017).
-
Tan, L.T.H.; Chan, K.G.; Khan T.M.; Bukhari, S.I.; Saokaew, S.; Duangjai, A.; Pusparajah, P.; Lee, L.H.; Goh, B.H. Streptomyces MUM212 as a source of antioxidants with radical scavenging and metal chelating properties. Front. Pharmacol. 2017, 8, 276. doi: 10.3389/fphar.2017.00276
-
Shanmugapriya, K.; Senthil, M.T.; Udayabhanu, J.; Thayumanavan, T. Antioxidant investigation of dried methanolic extracts of Gnaphalium polycaulon Pers, an Indian folkloric ethnomedicinal plant of the Nilgiri, Tamil Nadu, India. J. Phytomedicine. Clin. Ther. 2017, 5, 1. doi:10.21767/2321-2748.100317
Reviewer’s comment: 3
The discussion section is too extended and includes large portions that are “state of the art” and not discussion. This make the discussion difficult to follow
Author’s response: 3
The authors have removed an unnecessary part from the discussion section in the revised manuscript, as suggested by reviewer 2.
Reviewer’s comment: 4
What was the criteria for selecting the sequences from reference strains included in the phylogenetic tree? Information is missing.
Author’s response: 4
Thank you for the comments. The 16S rRNA gene sequence of DBT34 was compared with reference strains based EzTaxon database to see the homology similarity percentage of 96.5-99.0%. The highly closer reference sequence strains were selected and compared with our strains using Clustal-W analysis. After that, the phylogenetic tree was constructed using MEGA 6.0.
Reviewer’s comment: 5
It is said that “Among all of the tested isolates, Streptomyces sp. strain DBT34 again showed the maximum ability to scavenge free radicals, with a scavenging ability of 55.15%.”. However in the data that is presented 4 out of the 6 tested extracts showed higher activity.
Author’s response: 5
Thank you for comment. We agree with the reviewer comments. The authors have changed the sentences in the revised manuscript.
Round 2
Reviewer 1 Report
the manuscript was improved.
Author Response
Dear Reviewer
Thank you very much for your observations, we have re-checked the manuscript and highlighted the typo-errors as per your suggestions.
All corrections are incorporated in the manuscript and highlighted in yellow color.
Thank you again
Reviewer 2 Report
There is no evidence that functional protein (catalase or SOD) is present in the methanolic extracts. Unless a native-PAGE stained for catalase and/or SOD activity is presented, the authors should replace the expressions "catalase activity" and "sod activity" for "catalase-like activity" and "sod-like activity" throughtout the ms. Particularly in table 1.
Author Response
Dear Reviewer
Thank you for your suggestion, as per your suggestions we have replaced catalase and SOD activity with "catalase-like activity" and "sod-like activity" throughout the ms. Changed the same in Table 1 as suggested.
The part is mentioned in green color and highlighted in yellow.
Best regards